# Machine learning enables completely automatic tuning of a quantum device faster than human experts

H. Moon[1,8], D. T. Lennon [1,8], J. Kirkpatrick[2], N. M. van Esbroeck[1,3], L. C. Camenzind[4], Liuqi Yu[4], F. Vigneau [1], D. M. Zumbühl [4], G. A. D. Briggs [1], M. A. Osborne[5], D. Sejdinovic[6], E. A. Laird [7] & N. Ares [1✉]

Variability is a problem for the scalability of semiconductor quantum devices. The parameter space is large, and the operating range is small. Our statistical tuning algorithm searches for specific electron transport features in gate-defined quantum dot devices with a gate voltage space of up to eight dimensions. Starting from the full range of each gate voltage, our machine learning algorithm can tune each device to optimal performance in a median time of under 70 minutes. This performance surpassed our best human benchmark (although both human and machine performance can be improved). The algorithm is approximately 180 times faster than an automated random search of the parameter space, and is suitable for different material systems and device architectures. Our results yield a quantitative measurement of device variability, from one device to another and after thermal cycling. Our machine learning algorithm can be extended to higher dimensions and other technologies.

[1] Department of Materials, University of Oxford, Parks Road, Oxford OX1 3PH, UK. [2] DeepMind, London EC4 5TW, UK. [3] Department of Applied Physics, Eindhoven University of Technology, Eindhoven, MB 5600, The Netherlands. [4] Department of Physics, University of Basel, Basel 4056, Switzerland. [5] Department of Engineering, University of Oxford, Walton Well Road, Oxford OX2 6ED, UK. [6] Department of Statistics, University of Oxford, 24-29 St Giles, Oxford OX1 3LB, UK. [7] Department of Physics, Lancaster University, Lancaster LA1 4YB, UK. [8]These authors contributed equally: H. Moon, D.T. Lennon. ✉email: natalia.ares@materials.ox.ac.uk

Gate defined quantum dots are promising candidates for scalable quantum computation and simulation[1,2]. They can be completely controlled electrically and are more compact than superconducting qubit implementations[1]. These devices operate as transistors, in which electrons are controlled by applied gate voltages. If these gate voltages are set correctly, quantum dots are created, enabling single-electron control. If two such quantum dots are created in close proximity, the double quantum dot can be used to define robust spin qubits from the singlet and triplet states of two electrons[3,4]. Due to device variability, caused by charge traps and other device defects, the combination of gate voltage settings which defines a double quantum dot varies unpredictably from device to device, and even in the same device after a thermal cycle. This variability is one of the key challenges that must be overcome in order to create scalable quantum circuits for technological applications such as quantum computing. Typical devices require several gate electrodes, creating a high-dimensional parameter space difficult for humans to navigate. Tuning is thus a time-consuming activity and we are reaching the limits of our ability to do this manually in arrays of quantum devices. To find, in a multidimensional space, the gate voltages which render the device operational is referred to in the literature as coarse tuning[5,6].

Here, we present a statistical algorithm which is able to explore the entire multidimensional gate voltage space available for electrostatically defined double quantum dots, with the aim of automatically tuning them and studying their variability. Until this work, coarse tuning required manual input[7] or was restricted to a small gate voltage subspace[8]. We demonstrate a completely automated algorithm which is able to tune different devices with up to eight gate electrodes. This is a challenging endeavour because the desired transport features are only present in small regions of gate-voltage space. For most gate voltage settings, the device is either pinched off (meaning that the charge carriers are completely depleted so that no current flows) or too open (meaning that the tunnel barriers are too weakly defined for single-electron charge transport to occur). Moreover, the transport features that indicate the device is tuned as a double quantum dot are time-consuming to measure and difficult to parametrise. Machine learning techniques and other automated approaches have been used for tuning quantum devices[5–14]. These techniques are limited to small regions of the device parameter space or require information about the device characteristics. We believe our work significantly improves the state-of-the-art: our algorithm models the entire parameter space and tunes a device completely automatically (without human input), in approximately 70 min, faster than the typical tuning by a human expert.

Our algorithm explores the gate-voltage space by measuring the current flowing through the device, and its design makes only a few assumptions, allowing it to be readily applied to other device architectures. Our quantum dot devices are defined in a two-dimensional electron gas in a GaAs/AlGaAs heterostructure by Ti/Au gate electrodes. DC voltages applied to these gate electrodes, $V_1$–$V_8$, create a lateral confinement potential for electrons. Particularly important are the two plunger gate voltages $V_3$ and $V_7$, which mainly tune the electron occupation of the left and right dots. A bias voltage $V_{bias}$ is applied to ohmic contacts to drive a current ($I$) through the device. The device schematic, designed for precise control of the confinement potential[15–17], is shown in Fig. 1a. Measurements were performed at 50 mK.

We consider the space defined by up to eight gate voltages between 0 and $-2$ V. This range was chosen to avoid leakage currents. In this parameter space, the algorithm has to find the desirable transport features within tens of mV. Identifying these features is slow because it is requires measuring a two-dimensional current map, i.e., a plot of $I$ as a function of the two plunger gate voltages. Although other techniques for measuring the double quantum dot exist, such as charge sensing and dispersive readout, they also require other parameters to be retuned when the gate voltages vary and are therefore not suitable for automated measurements. Our algorithm is thus designed to minimize the number of current maps that it requires to find the transport features in question.

We make two observations. Firstly, that for very negative gate voltages, no current will flow through the device, i.e., the device is pinched-off. Conversely, for very positive gate voltages, full current will flow and single electron transport will not be achieved. This means that transport features are expected to be found near the hypersurface that separates low and high current regions in parameter space. The second observation is that to achieve single-electron transport, a confinement potential is needed. The particular transport features that evidence single-electron transport are Coulomb peaks, which are peaks in the current flowing through the device as a function of a single plunger gate voltage. These observations lead us to only two modelling assumptions: (i) single and double quantum dot transport features are embedded near a boundary hypersurface, shown in Fig. 1b, which separates regions in which a measurable current flows, from regions in which the current vanishes; (ii) large regions of this hypersurface do not display transport features.

The algorithm consists of two parts: a sampling stage that generates candidate locations on the hypersurface, and an investigation stage in which we collect data in the vicinity of each candidate location, i.e., close to the candidate location in gate voltage space (see Section "Investigating nearby voltage space, for precise definitions of the size of the regions explored around candidate locations"), to evaluate transport features (Fig. 2). The results of the investigation stage feed back into the sampler, which chooses a new candidate location in the light of this information. The purpose of the sampler is to produce candidate locations in gate voltage space for which the device operates as a double quantum dot. A block diagram of the algorithm is displayed in Fig. 3. Our modelling assumptions are based on the physics of gate defined devices leading to minimal constraints; we do not assume a particular shape for the hypersurface, and we instead allow measurements to define it by fitting the data with a Gaussian process. Overall, the algorithm minimises tuning times by identifying candidate locations on a hypersurface model that is updated with each measurement; by prioritising the most promising of these locations; and by avoiding the acquisition of two-dimensional current maps which do not correspond to a double quantum dot regime.

We demonstrate over several runs, in two different devices and over multiple thermal cycles, that the algorithm successfully finds transport features corresponding to double quantum dots. We perform an ablation study, which identifies the relative contribution of each of the modules that constitute the algorithm, justifying its design. Finally, we demonstrate that our algorithm is capable of quantifying device variability, which has only been theoretically explored so far[18]. We have done this by comparing the hypersurfaces found for different devices and for a single device in different thermal cycles.

Automating experimental science has the impact to significantly accelerate the process of discovery. In this work we show that a combination of simple physical principles and flexible probabilistic machine learning models can be used to efficiently characterise and tune a device. We envisage that in the near future such judicious application of machine learning will have tremendous impact even in areas where only small amounts of data are available and no clear fitness functions can be defined.

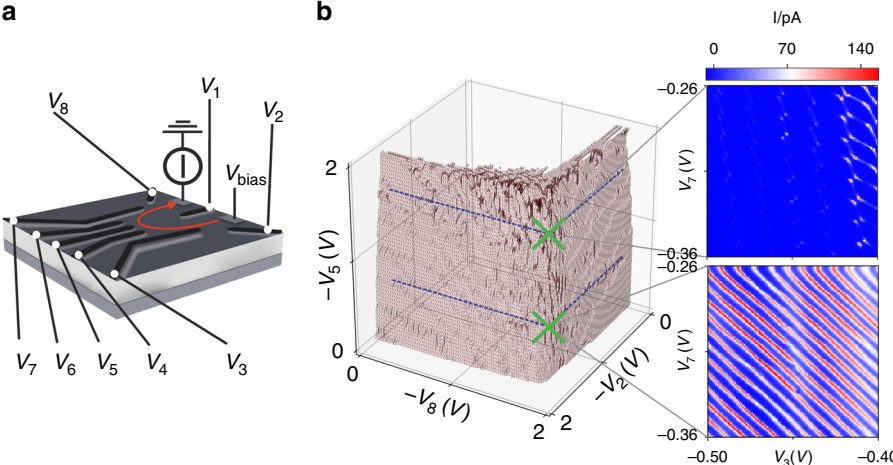

**Fig. 1 Overview of device, and gate voltage space. a** Schematic of a gate-defined double quantum dot device. **b** Left: Boundary hypersurface measured as a function of $V_2$, $V_5$, and $V_8$, with fixed values of $V_1$, $V_3$, $V_4$, $V_6$, and $V_7$. The current threshold considered to define this hypersurface is 20% of the maximum measured current. The gate voltage parameter space, restricted to 3D for illustration, contains small regions in which double and single quantum dot transport features can be found. These regions typically appear darker in this representation because they produce complex boundaries. Right: For particular gate voltage locations marked with green crosses, the current as a function of $V_7$ and $V_3$ is displayed. The top and bottom current maps display double and single quantum dot transport features, respectively.

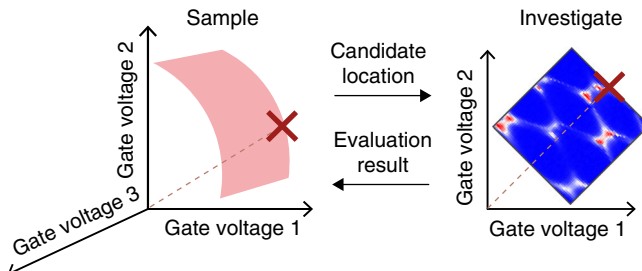

**Fig. 2 Overview of the algorithm.** The sampling phase stage produces candidate locations in gate voltage space, which are on the boundary hypersurface (pink surface). The distance between a candidate location (red cross) and the origin of the gate voltage space is marked with a dashed line. The investigation stage evaluates the local region by, for example, measuring current maps which are evaluated by a score function. (The current map displayed is an example of a measurement performed by the algorithm. It uses a colour scale running from red, the highest current measured, to blue, the lowest current). Evaluation results are fed back to the sampling stage.

## Results

**Description of the algorithm**. The algorithm starts with an initialization stage. This stage begins with setting $V_{bias}$. The current is then measured at the two extremes of the gate voltage space, $V_j = 0$ and $V_j = -2$ V for $j = 1, \ldots, N$, where $N$ is the number of gate electrodes. For the most negative extreme, the measured current should be 0, but current offsets might change this value for different measurement setups. The difference between the currents at these two extremes is the full-scale current which is used to set the threshold that defines the hypersurface. The search range was chosen as the typical gate voltage range used when tuning similar devices from scratch.

The algorithm then begins an iterative process during which it alternates between the sampling and investigation stages. In each iteration, the sampling stage identifies a candidate location on the hypersurface in voltage space, attempting to select locations with a high probability of desirable transport features. The investigation stage then explores the nearby region of voltage space, attempting to identify whether current maps measured in this region show Coulomb peaks and honeycomb patterns. The presence of Coulomb peaks is reported back to the sampling stage as an evaluation result, which it uses in future iterations to inform its selection of new candidates. The steps that make up each iteration will now be described in detail.

*Searching for the hypersurface.* In each iteration, the algorithm first locates the hypersurface in gate voltage space. To do this, it selects a search direction, specified by a unit vector **u** which during the first 30 iterations of the algorithm is selected randomly from a hypersphere, restricted to the octant where all gate voltages are negative. The gate voltages are then scanned along a ray beginning at the origin **o** and parallel to **u** (Fig. 4a). During this scan, the current is monitored; when it falls below a threshold of 20% of full scale, this is taken as defining a location **v(u)** on the hypersurface.

While this procedure correctly identifies locations for which current through the device is pinched off, it does not recognise whether the device is "tunable" in the sense that every gate voltage strongly affects the current. We find that for some locations, most gate voltages have little effect, which suggests that the measured current is not being determined by the potential in the quantum dot. With such a combination of gate voltages, a double quantum dot cannot be usefully formed. To reduce the amount of time spent exploring such regions of the hypersurface, we implemented the following heuristic pruning process (Fig. 4b), applied in each of the first 30 iterations. From the hypersurface intersection **v(u)**, all voltages are stepped upwards to a location $\mathbf{v}^{\delta}(\mathbf{u}) \equiv \mathbf{v}(\mathbf{u}) + \boldsymbol{\delta}$, where $\boldsymbol{\delta}$ is a step-back vector with each component chosen to be +100 mV. Each voltage in turn is then swept downwards towards the bottom of its range or until the hypersurface is encountered. If the hypersurface is encountered only along one voltage axis $k$, then the origin for subsequent iterations is moved so that its $k$-component is equal to the $k$-component of $\mathbf{v}^{\delta}$ (Fig. 4b, inset). Over several iterations, this process prunes away search paths for which the hypersurface is not intersected within the chosen range.

*Investigating nearby voltage space.* Having located the hypersurface, the algorithm then proceeds to investigate the nearby region of voltage space to determine whether a double quantum dot is

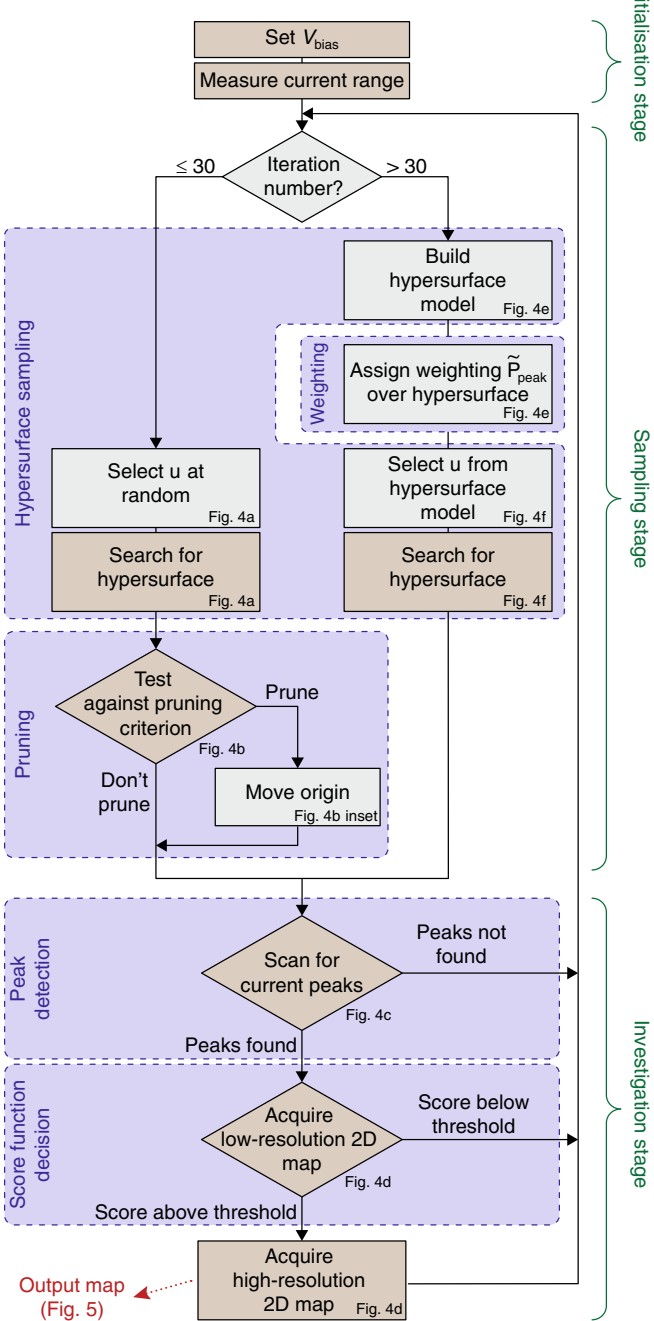

**Fig. 3 Flow diagram of the algorithm.** (See text and Fig. 4 for a full description.) Each step is annotated with the corresponding panel in Fig. 4. Steps that involve interaction with the device are shaded brown, and entirely computational steps are grey. In the ablation studies of Section "Ablation study", the different modules that constitute the algorithm are studied in terms of their contribution to the algorithm's performance; these modules are marked by the blue background regions. The steps belonging to the initialisation, sampling, and investigation steps are indicated on the right.

formed. The investigation is carried out in the plane containing $\mathbf{v}(\mathbf{u})$ and defined by varying the two plunger gate voltages $V_3$ and $V_7$. These gates, selected before running the algorithm, are the ones that should predominantly shift the electrochemical potential in the left and right dots. If a double quantum dot is formed, the current should therefore show a honeycomb pattern in this plane, similar to Fig. 2.

First, a one-dimensional scan is made in this investigation plane, starting at $\mathbf{v}(\mathbf{u})$ and running along the diagonal axis $\hat{V}_e \equiv \frac{1}{\sqrt{2}}(\hat{V}_3 + \hat{V}_7)$, where $\hat{V}_i$ indicates a unit vector in voltage space (Fig. 4c). This scan is chosen to have length 128 mV and resolution 1 mV. A peak detection routine identifies the presence or absence of Coulomb peaks. If Coulomb peaks are absent, investigation here ceases and a new iteration begins.

Next, if Coulomb peaks are present in this diagonal scan a two-dimensional scan is made (Fig. 4d). The scanning region is a square oriented along $\hat{V}_e$ and its orthogonal axis $\hat{V}_a \equiv \frac{1}{\sqrt{2}}(\hat{V}_3 - \hat{V}_7)$. This square is bounded by $\mathbf{v}(\mathbf{u})$, and its side length is chosen to be 3.5 times the average peak spacing identified in the diagonal scan. (If the diagonal scan shows less than 3 peaks, the side length is set to be 100 mV.) The scan is made first at low resolution ($16 \times 16$ pixels), and a score is assigned to the resulting current map. The score function (see Supplementary Methods, Score function) is a predefined mathematical expression designed to reward specific transport features that correspond to the visual features typically looked for by humans when manually tuning a device. In particular, it is designed to identify honeycomb patterns similar to Fig. 2 indicating the formation of a double quantum dot. It rewards current maps containing sharp and curved lines.

If the score function of the low-resolution scan is high, it is repeated at high resolution ($48 \times 48$ pixels). The score threshold is dynamically adjusted throughout the experiment so that 15% of low-resolution scans are repeated. (See Supplementary Methods, Optimal threshold $\alpha'$, for a statistical analysis of the optimal threshold.) The high-resolution maps, scanned in regions of voltage space identified as showing desirable double-dot behaviour, constitute the output of the tuning algorithm.

*Searching efficiently by learning about the hypersurface.* To more rapidly locate the hypersurface, and to increase the fraction of time spent exploring regions of gate space containing Coulomb peaks, the algorithm improves the search process of Section Searching for the hypersurface by incorporating information from its measurements. It applies this information beginning with the 31st iteration. To do this, it starts each iteration by using the measured locations of the hypersurface to generate a model hypersurface spanning the entire voltage space (Fig. 4e). The model is generated using a Gaussian process[19] incorporating the uncertainty of the measured locations as explained in the Supplementary Methods, Gaussian process models. To each candidate search direction $\mathbf{u}$, the model assigns an estimated distance to the hypersurface $m(\mathbf{u})$ with uncertainty $s(\mathbf{u})$. Furthermore, the model uses information on whether current peaks were identified in previous searches to assign to each point on the model hypersurface a probability $\tilde{P}_{\text{peak}}$ of expecting peaks.

Using this model, the algorithm can now select new search directions $\mathbf{u}$ more efficiently. It is desirable to select search directions associated with a high probability $\tilde{P}_{\text{peak}}$, while also occasionally exploring less promising regions of the hypersurface. To achieve this trade-off, the algorithm first generates a set of candidate search locations on the hypersurface (Fig. 4f). To generate a set that is approximately uniform despite the convoluted shape of the hypersurface, we adopt a selection routine based on simulated Brownian motion[20]; a set of "particles" is simulated inside the hypersurface, and each encounter with the hypersurface contributes one candidate location (Fig. 4f, inset). To each of these locations, the algorithm then assigns a weight proportional to the corresponding value of $\tilde{P}_{\text{peak}}$, and selects one location at random (i.e., using Thompson sampling). This location then defines a new search direction $\mathbf{u}$.

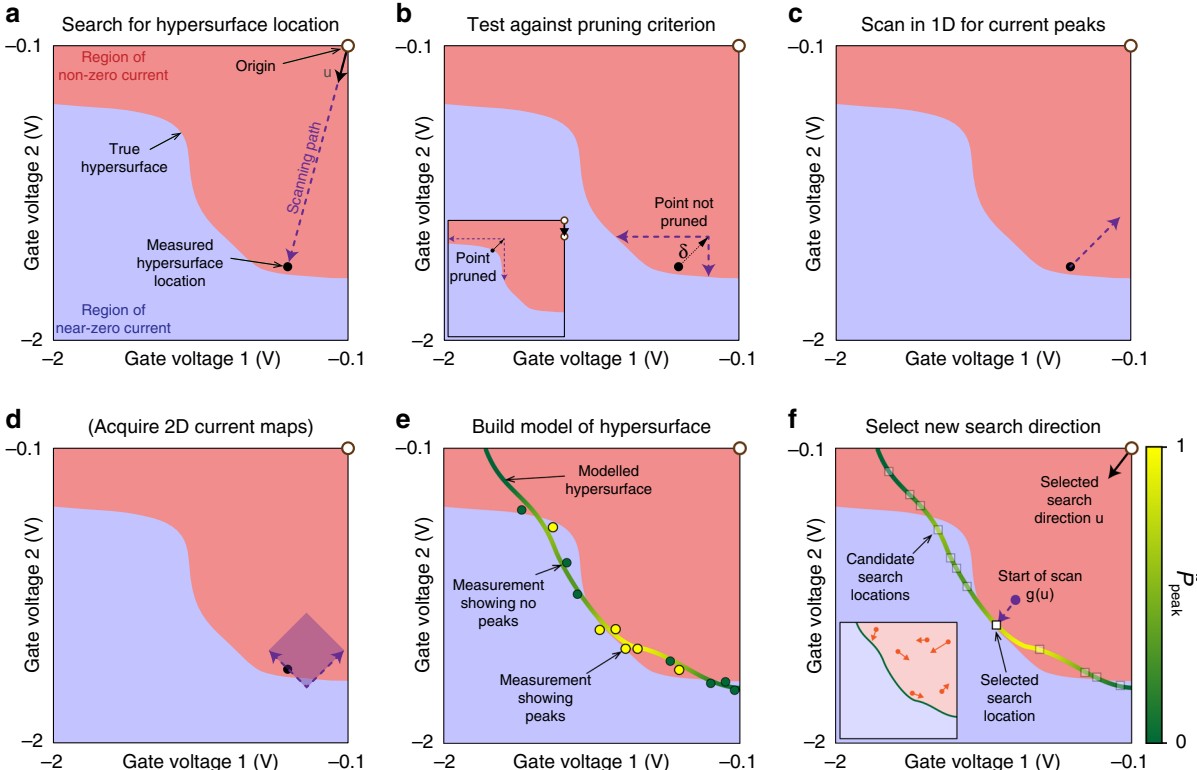

**Fig. 4 Characterising the boundary hypersurface using machine learning.** Each panel illustrates a step of the algorithm presented in Fig. 3. The gate voltage space, restricted to two dimensions for illustration, is divided into regions of near-zero (blue) and non-zero current (pink), separated by a boundary hypersurface. **a** Locating the hypersurface. The gate voltages are scanned along a ray (violet arrow) starting at the origin (white circle) and defined by direction **u**. By monitoring the current, the intersection with the hypersurface is measured. **b** To determine whether a region should be pruned, the algorithm scans each gate voltage individually toward the bottom of its range from a location just inside the hypersurface as shown. If only one scan intersects the hypersurface (as in the inset), future exploration of that region is inhibited by displacing the origin as shown. **c** Based on a short 1D scan, the location is classified according to whether it shows current peaks indicating Coulomb blockade. **d** If peaks are found, a 2D scan (violet square) is performed in the plane of $V_3$ and $V_7$, and is possibly repeated at higher resolution. **e** From the first thirty measurements (green and yellow circles), the algorithm builds a model of the hypersurface and assigns a probability $\tilde{P}_{peak}$ that peaks will be found. **f** To refine the model, the algorithm generates a set of candidate search locations (squares), each weighted by its corresponding value of $\tilde{P}_{peak}$, and selects one at random. A new scan is then performed in the corresponding direction to generate a new measurement of the hypersurface location. Steps **d**–**f** are then repeated indefinitely. Inset: Scheme for numerically sampling the hypersurface using simulated Brownian motion. Each point represents a simulated particle moving inside the enclosed volume. The collisions between the particles and the modelled hypersurface generate a set of candidate search locations.

The model hypersurface is also used to improve the efficiency of the search. Instead of beginning at the origin (as in Fig. 4a), the new search scan begins at the location $\mathbf{g}(\mathbf{u}) \equiv \mathbf{o} + (m(\mathbf{u}) - 2s(\mathbf{u}))$ $\mathbf{u}$, which should lie just inside the hypersurface (as in Fig. 4f). If $m(\mathbf{u}) - 2s(\mathbf{u}) < 0$, the search scan begins at $\mathbf{o}$.

Occasionally, the measured current at the beginning of the scan is below threshold, indicating that $\mathbf{g}(\mathbf{u})$ is already in the pinched-off region. In these cases, the algorithm scans in the opposite direction, along—$\mathbf{u}$. Once the measured current increases above 0.8 of the value at $\mathbf{o}$, the algorithm reverts to measuring in the $\mathbf{u}$ direction to locate the hypersurface in the usual way. Over many iterations, the algorithm thus builds up the required set of high-resolution current maps, measured with constantly improved efficiency.

**Experimental results**. The performance of our algorithm is assessed by a statistical analysis of the expected success time $\mu_t$. This is defined as the time it takes the algorithm to acquire a high-resolution current map that is confirmed a posteriori by humans as containing double quantum dot features. Note that this confirmation is only needed to assess the performance of the algorithm. Because human labelling is subjective, three different researchers labelled all current maps, deciding in each case if they could identify features corresponding to the double quantum dot

regime, with no other information available. See Supplementary Methods, Bayesian statistics, for details of the multilabeller statistical analysis.

*Device tuning*. To benchmark the tuning speed of our algorithm, we ran it several times on two different devices with identical gate architecture, Devices 1 and 2, and we compared its performance with a Pure random algorithm. The Pure random algorithm searches the whole gate voltage parameter space by producing a uniform distribution of candidate locations. Unlike our algorithm, which we will call Full decision, it does not include hypersurface weighting or pruning rules, but uses peak detection in its investigation stage. All Full decision runs presented in this section for Device 1 and Device 2 were performed during a single cool down (cool down 1). The Pure random runs in each device were performed in a different cool down (cool down 2).

As mentioned in the introduction, we consider a gate voltage space whose dimension is defined by the number of working gate electrodes, and we provide a gate voltage range that avoids leakage currents. While for Device 1 we considered the eight-dimensional parameter space defined by all its gate electrodes, for Device 2 we excluded gate electrode 6 by setting $V_6 = 0$ mV due to observed leakage currents associated with this gate.

We define the average count $\bar{C}$ as the number of current maps labelled by humans as displaying double quantum dot features divided by the number of labellers. For a run of the Pure random algorithm in Device 2 and five runs of our algorithm in Devices 1 and 2, we calculated $\bar{C}$ as a function of laboratory time (Fig. 5a, b). We observe that $\bar{C}$ is vastly superior for our algorithm compared with Pure random, illustrating the magnitude of the parameter space.

The labellers considered a total of 2048 current maps produced in different runs, including those of the ablation study in Section Ablation study. The labellers had no information of the run in which each current map was produced, the device or the algorithm used. For the Pure random approach, the labelled set was composed of 51 current maps produced by the algorithm and 100 randomly selected from the set of 2048.

The time $\mu_t$ is estimated by the multi-labeller statistics. The multi-labeller statistics uses an average likelihood of $\mu_t$ over multiple labellers and produces an aggregated posterior distribution (see Supplementary Methods, Bayesian statistics). From this distribution, the median and 80% (equal-tailed) credible interval of $\mu_t$ is 2.8 h and (1.9, 7.3) h for Device 1 and 1.1 h and (0.9, 1.6) h for Device 2. Experienced humans require approximately 3 h to tune a device of similar characteristics into exhibiting double quantum dot features (F. Kuemmeth, personal communication). Our algorithm's performance might therefore be considered super human. Due to device variability, the hypersurfaces of these two devices are significantly different, showing our algorithm is capable of coping with those differences.

In Fig. 5c, d, we compare the probability of measuring Coulomb peaks in the vicinity of a given $\mathbf{v}(\mathbf{u})$, $P$(peaks), for Pure random and different runs of our algorithm. We calculate $P$(peaks) as the number of sampled locations in the vicinity of which Coulomb peaks were detected over $n$. In this way, we confirm that $P$(peaks) is significantly increased by our algorithm. It has a rapid growth followed by saturation. Fig. 5e, f shows the high resolution current maps produced for Device 2 by Pure random and one of our algorithm runs. We observe that our algorithm produces high resolution current maps which are recognized by all labellers as displaying double quantum dot features within 1.53 h. The three current maps in Fig. 5f correspond to double quantum dot regimes found by our algorithm in different regions of the gate voltage space. The number of labellers $C$ who identify the current maps produced by Pure random as corresponding to double quantum dots, $C$, is 0 or 1. This demonstrates our algorithm finds double quantum dot regimes, which can be later fine tuned to reach optimal operation conditions fully automatically[21].

To significantly reduce tuning times, we then modified our algorithm to group gate electrodes that perform similar functions. The algorithm assigns equal gate voltages to gate electrodes in the same group. For Device 1, we organized the eight gate electrodes into four groups: $G_1 = (V_1)$, $G_2 = (V_2, V_8)$, $G_3 = (V_3, V_7)$, and $G_4 = (V_4 - V_6)$. In this case, the median and 80% credible interval of $\mu_t$ improve to 0.6 h and (0.4, 1.1) h (see Supplementary Fig. 2 for a plot of $\bar{C}$). This approach, by exploiting knowledge of the device architecture, reduces $\mu_t$ by more than four times.

*Ablation study.* Our algorithm combines a sampling stage, which integrates the hypersurface sampling with weighting and pruning, and an investigation stage that includes peak detection and score function decisions. Each of these modules, illustrated in Fig. 3, contributes to the algorithm's performance. An ablation study identifies the relative contributions of each module, justifying the algorithm's architecture. For this ablation study we chose to compare our algorithm, Full decision, with three reduced versions that combine different modules; Pure random, uniform surface, and peak weighting (see Table 1).

Pure random, defined in the previous section, produces a uniform distribution of candidate locations over the whole gate voltage space. It excludes the sampling and pruning rules. Uniform surface makes use of the hypersurface sampling, but no weighting or pruning rules are considered. Peak weighting combines the hypersurface sampling with weighting and pruning rules. These three algorithms use peak detection in their investigation stage, but none of them use the score function decision. For the ablation study, we define low (high) resolution as $20 \times 20$ ($60 \times 60$) pixels.

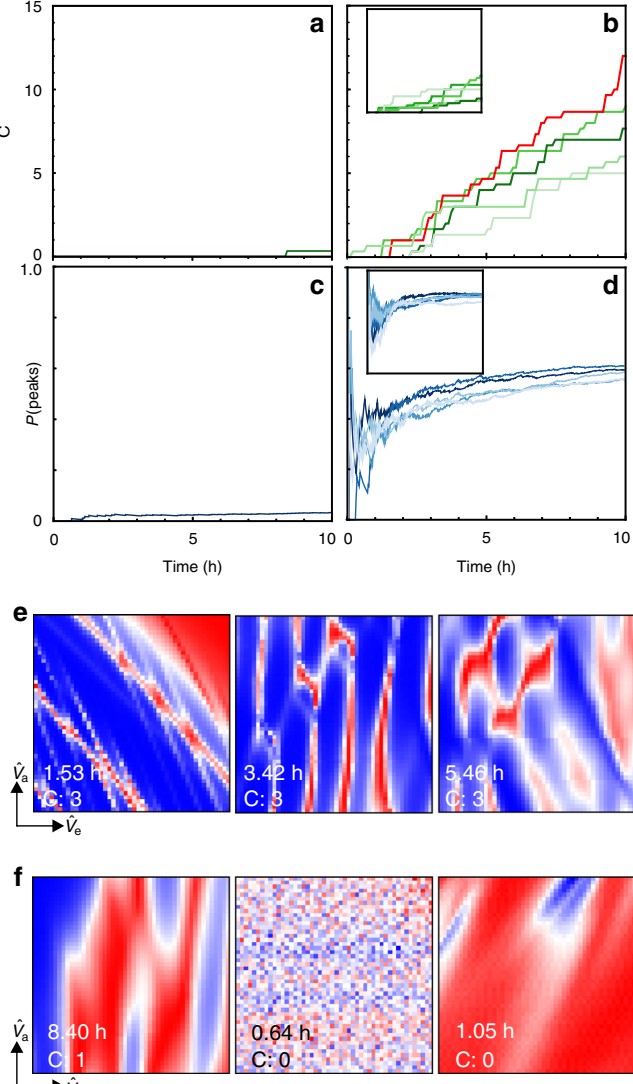

**Fig. 5 Algorithm's performance. a–d** Average number of current maps displaying double quantum dot features, $\bar{C}$, and $P$(peaks) as a function of laboratory time. Current maps are labelled by humans a posteriori, i.e., after the algorithm is stopped. **a**, **c**, **b**, **d** correspond to one run of Pure random and five runs of our algorithm, respectively. All algorithm runs displayed in main panels were performed in Device 2, while insets show runs of our algorithm in Device 1. **e**, **f** High resolution current maps measured in Device 2 by Pure random and one of our algorithm runs, respectively. We indicate the time the algorithm had been running for before they were acquired and the number of labellers, $C$, that identified them as displaying double quantum dot features. Current maps are ordered from left to right in decreasing order of $C$, and maps that have the same values of $C$ are displayed in the order at which they were sampled. Each panel uses an independent colour scale running from red (highest current measured) to blue (lowest current).

| Table 1 Comparison of algorithms used in the ablation study. | | | | |
|---|---|---|---|---|
| **Algorithm** | **SS: Hypersurface sampling** | **SS: Weighting and pruning** | **IS: Peak detection** | **IS: Score function decision** |
| Pure random | × | × | ✓ | × |
| Uniform surface | ✓ | × | ✓ | × |
| Peak weighting | ✓ | ✓ | ✓ | × |
| Full decision | ✓ | ✓ | ✓ | ✓ |
| Modules in the sampling stage (SS) and the investigation stage (IS) are indicated with a tick if included and with a cross if excluded. | | | | |

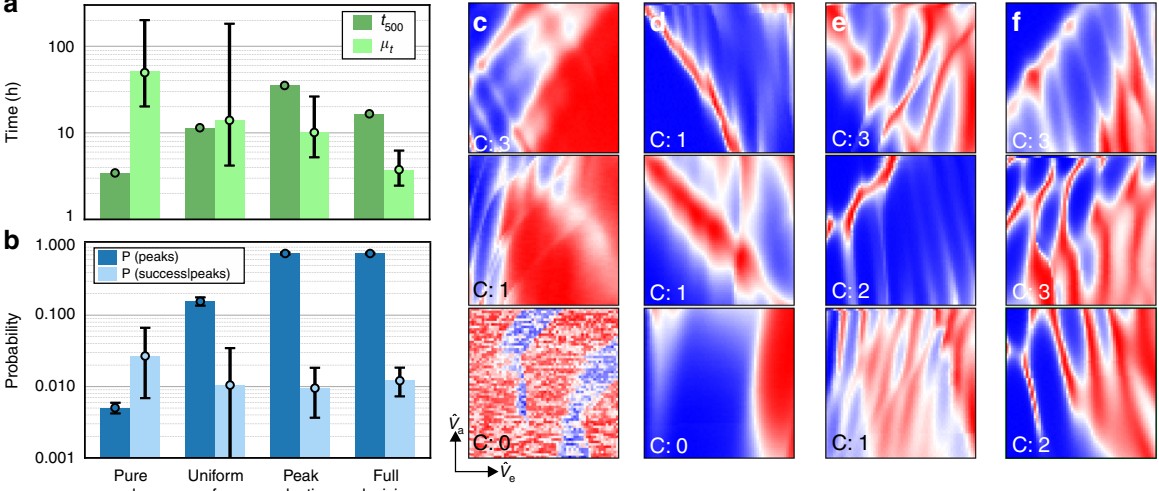

**Fig. 6 Ablation study. a, b** Bar charts and corresponding data points comparing $\mu_t$ (light green), $t_{500}$ (dark green), $P$(peaks) (dark blue) and $P$(success|peaks) (light blue) for the different algorithms considered. Error bars represent 80% (equal-tailed) credible intervals. Due to a measurement problem, 459 sampling iterations instead of 500 were considered for the Full decision algorithm. **c–f** High resolution current maps sampled by pure random, uniform surface, peak weighting, and full decision, respectively. In each panel, we indicate $C$, the number of human labellers that identified a map as displaying double quantum dot features. Current maps with identical values of $C$ are displayed in the order in which they were sampled, from top to bottom. Current maps with $C$: 0 were randomly selected. Each panel uses an independent colour scale running from red (highest current measured) to blue (lowest current).

To analyse the algorithm's performance, we estimate $P$(peaks) and the probability of success, i.e., the probability to acquire a high-resolution current map labelled as containing double quantum dot features, given Coulomb peak measurements $P$(success|peaks). To take measurement times into consideration, we define $t_{500}$ as the time to sample and investigate 500 locations in gate voltage space. The ablation study was performed in Device 1 keeping investigation stage parameters fixed. The cool down cycle was the same as in Section "Device tuning" (cool down 1), except for Pure random, which was performed in a new thermal cycle (cool down 2). Results are displayed in Fig. 6.

Figure 6a shows that the introduction of the hypersurface sampling, and weighting and pruning, increases $t_{500}$. This is because $P$(peaks) increases with these modules (Fig. 6b), and thus the number of low and high resolution current maps required by the investigation stage is larger. Within uncertainty, $P$(success|peaks) remains mostly constant for the different algorithms considered. The result is a decreasing $\mu_t$ from Pure random to Peak weighting within experimental uncertainty. See "Methods", "Mathematical analysis of ablation study results, for a mathematical analysis of these results".

The reason behind the use of peak detection in all the algorithms considered for this ablation study is the vast amount of measurement time that would have been required otherwise. Without peak detection, the posterior median estimate of $\mu_t$ for Pure random is 680 h.

To complete the ablation study, we compare the considered algorithms with the grouped gates approach described in the previous section, keeping parameters such as the current map resolutions are equal. We found $\mu_t = 80.5$ min (see Supplementary Fig. 3 for a plot comparing these algorithms).

In summary, comparing Pure random and Uniform surface, we show the importance of hypersurface sampling. The difference between Uniform surface and Peak weighting highlights the importance of weighting and pruning. The improved performance of Full decision with respect to Peak weighting evidences the tuning speedup achieved by the introduction of the score function. These results demonstrate Full decision exhibits the shortest $\mu_t$ and imply an improvement over Pure random without peak detection of approximately 180 times.

**Device variability**. The variability of electrostatically defined quantum devices has not been quantitatively studied so far. We have been able to exploit our algorithms for this purpose. Using the uniform surface algorithm only (no investigation stage), we obtain a set of locations on the hypersurface $\mathbf{v}_a$. Changes occurring to this hypersurface are detected by running the algorithm again and comparing the new set of locations, $\mathbf{v}_b$, with $\mathbf{v}_a$. This comparison can be done by a point set registration method, which allows us to find a transformation between point sets, i.e., between the hypersurface locations.

Affine transformations have proven adequate to find useful combinations of gate voltages for device tuning[9,10]. To find a measure of device variability, understood as changes occurring to a device's hypersurface, we thus use an affine transformation $\mathbf{v}_t = B\mathbf{v}_b$, with $B$ a matrix which is a function of the transformation's parameters. We are looking for a transformation of coordinates

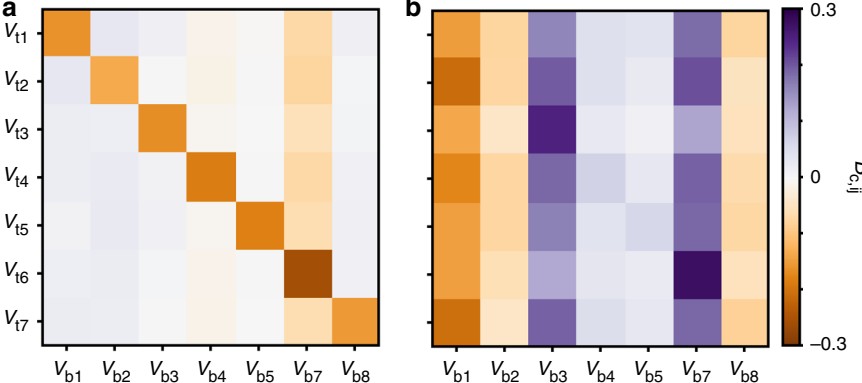

**Fig. 7 Learning about device variability.** $B_c$ matrices obtained using point set registration. Indices are the gate voltage locations $\mathbf{v}_b$ and $\mathbf{v}_t$. $V_6 = 0$ mV was fixed in Device 2 to prevent leakage currents. **a** Transformation between the hypersurface of Device 2 before and after a thermal cycle. **b** Transformation between the hyperfsurfaces of Device 1 and Device 2.

that converts $\mathbf{v}_b$ into a set of locations $\mathbf{v}_t$ which is as similar to $\mathbf{v}_a$ as possible.

The particular point set registration method we used is coherent point drift registration[22]. This method works with an affine transformation which includes a translation vector. We have modified the method to set this translation vector to zero, as the transformation between hypersurfaces can be fully characterized by the matrix $B$ (see Supplementary Methods, Point set registration).

We have used this approach to quantify the variability between Devices 1 and 2, and the effect of a thermal cycle in the hypersurface of Device 2. Figure 7 displays the matrix $B_c = B - I$ for each case, quantifying how much $B$, the transformation that converts a set of locations from one hypersurface onto the other, differs from the identity matrix ($I$). Nonzero elements of $B_c$ thus indicate device variability. Diagonal elements of $B_c$ are responsible for scale transformations and can be interpreted as a capacitance change for a given gate electrode. Off-diagonal elements are responsible for shear transformations and can be interpreted as a change in cross-capacitance between a pair of gate electrodes.

Figure 7a shows $B_c$ corresponding to the changes in the hypersurface of Device 2 after a thermal cycle (cool down 1 vs. cool down 3). This transformation shows that device variability in a thermal cycle is dominated by a uniform change in capacitance for all gate electrodes. We have also measured $B_c$ for a thermal cycle of Device 1 (see Supplementary Fig. 4). Figure 7b displays $B_c$ comparing the hypersurface of Device 1 (cool down 1) with the hypersurface of Device 2 (cool down 3). We observe that the variability between these devices, which share a similar gate architecture, is given by nonuniform changes in gate electrode capacitance, as well as by changes in cross-capacitance. This variability is attributed to charge traps and other device defects, such as a small differences in the patterning of gate electrodes.

## Discussion

We demonstrated an algorithm capable of tuning a quantum device with multiple gate electrodes in approximately 70 min. This was achieved by efficiently navigating a multidimensional parameter space without manual input or previous knowledge about the device architecture. This tuning time was reproduced in different runs of the algorithm, and in a different device with a similar gate architecture. Our tuning algorithm is able to tune devices with different number of gate electrodes with no modifications. We showed that gate electrodes with similar functions can be grouped to reduce the dimensionality of the gate voltage space and reduce tuning times to 36 min. Tuning times might be further improved with efficient measurement techniques[23], as

measurement and gate voltage ramping times were found to be the limiting factor. The use of charge sensors and RF readout could also be implemented to improve tuning times, although these techniques would require to be automatically tuned to their optimum operating configuration, and would be restricted to small regions of the gate voltage space. We analysed our algorithm design through an ablation study, which allowed us to justify and highlight the importance of each of its modules. The improvement over the pure random search without peak detection is estimated to be 179 times.

We showed that device variability can be quantified using point set registration by uniform sampling of the hypersurface separating regions of high and low current in gate voltage space. We found that variability between devices with similar gate architectures is given by nonuniform changes in gate capacitances and cross-capacitances. Variability across thermal cycles is only given by a uniform change in gate capacitances.

Other device architectures might use the sampling stage of our algorithm as a first tuning step, and the investigation stage can be adapted to tune quantum devices into more diverse configurations. To achieve full automated tuning of a singlet–triplet qubit, it will be necessary to go beyond this work by tuning the quantum dot tunnel barriers, identifying spin-selective transitions, and configuring the device for single-shot readout.

## Methods

**The score function as a classifier.** One of key strength of the proposed algorithm is that it does not require an ideal score function. It is important to highlight that we are using the score function just as a classifier, instead of aiming at finding the gate voltage configuration that maximises the score. The reason for this is threefold; (i) the score function is not always a smooth function; (ii) it does not always capture the quality of the transport features; (iii) it is just designed for a particular transport regime, in this case, honeycomb patterns. Therefore, the score threshold acts as a parameter that just controls the characteristics of the classifier. If the threshold is low, many high resolution scans not leading to double quantum dot transport features are produced. If the threshold is too high, then promising gate voltage windows are missed. The optimal threshold can be estimated by minimising the time required to produce a high-resolution current map that is labelled by humans as containing double quantum dot features.

**Mathematical analysis of ablation study results.** The results in the ablation study can be verified under a few assumptions by a mathematical derivation of $\mu_t$ (see Supplementary Methods, Mathematical derivation of $\mu_t$). From this derivation, we can compare the expected times $\mu_t^{\text{abl}}$ for Pure random, Uniform surface, and Peak weighting:

$$\mu_t^{\text{abl}} = \frac{\mu_i^{\text{abl}}}{P(\text{success})}, \tag{1}$$

where $\mu_i^{\text{abl}}$ is the expected time per each iteration of the algorithm, and $P(\text{success}) = P(\text{peaks})P(\text{success}|\text{peaks})$ is the probability that double quantum dot transport features are observed in a high resolution scan at a given iteration of the algorithm.

For each iteration, time is required for a low resolution scan $t_{2D-L}$, a high resolution scan $t_{2D-H}$, and for the rest of the investigation and sampling $t_{others}$, including ramping gate voltages, peak detection, and computation time. The simulation of the Brownian particles is conducted in parallel with the investigation stage of the location proposed by the sampler in a previous run, and it does not increase $t_{others}$. As a result, the expected time for each iteration is

$$\mu_i^{abl} = t_{others} + P(peaks)t_{2D}, \qquad (2)$$

where $t_{2D} = t_{2D-L} + t_{2D-H}$. Note that 2D scans are acquired with probability $P(peaks)$. If the score function decision is not included, high resolution current maps are always acquired when Coulomb peaks are detected. In this case, low resolution current maps are not useful, but we have still included $t_{2D-L}$ in $t_{2D}$ to keep the comparison between algorithms consistent.

For all methods in Table 1 except Pure random, the time for 2D scans is the same, $t_{2D-L} \approx 33$ s and $t_{2D-H} \approx 273$ s, and $t_{others} \approx 35$ s. Therefore, the difference on $\mu_i^{abl}$ across methods is given by $P(peaks)$ and $P(success|peaks)$. In Fig. 6b, we can see that $P(success|peaks)$ is similar across the different algorithms, but $P(peaks)$ is different. In conclusion, $P(peaks)$ in Eqs. ((1)) and ((2)) determines $t_{500}$ and $\mu_t$ in Fig. 6a.

Rearranging $\mu_t^{abl}$ yields

$$\mu_t^{abl} = \left(\frac{t_{others}}{P(peaks)} + t_{2D}\right)\frac{1}{P(success|peaks)},$$

and this implies that $t_{2D}$ has a significant weight when $P(peaks)$ is large, motivating the introduction of the score function.

The expected time for Full decision algorithm is

$$\mu_i^{full} = t_{others} + P(peaks)t_{2D-L} + P(highres)t_{2D-H}$$
$$\mu_t^{full} = \frac{\mu_i^{full}}{P(success)},$$

where $P(highres)$ is the probability of acquiring a high resolution current map given a score. The score function decision always makes $\mu_t^{full}$ smaller than $\mu_t^{abl}$, because $\mu_t^{abl} - \mu_t^{full} = P(peaks)(1 - P(highres|peaks))t_{2D-H}$ and $P(highres|peaks)$ < 1. This is experimentally verified in Fig. 6 from the fact that $t_{500}$ of Full decision is smaller than that of Peak weighting.

Comparisons between $\mu_t^{abl}$ and $\mu_t^{full}$ can be affected by the dependence of $P$ $(success|peaks)$ on the score function threshold. In Fig. 6b, however, we observe that $P(success|peaks)$ is similar for Peak weighting and Full decision. This implies that the introduction of a score function threshold does not reduce the probability of success.

In this case,

$$\mu_t^{abl} - \mu_t^{full} = \frac{1 - P(highres|peaks)}{P(success|peaks)}t_{2D-H}.$$

This equation confirms that that the score function reduces $\mu_t$ in the case that the score function threshold does not degrade $P(success|peaks)$. Further analysis on the optimal threshold, i.e, the threshold that minimizes $\mu_t^{full}$, can be found in Supplementary Methods, Optimal threshold $\alpha'$.

## Data availability
The data acquired by the algorithm during experiments is available from the corresponding author upon reasonable request.

## Code availability
The original implementation of the algorithm and a refactored version (with examples and documentation) that is easier to deploy are available. Original version: https://doi.org/10.5281/zenodo.3966350. Refactored version: https://doi.org/10.5281/zenodo.3966318.

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

## Acknowledgements
We acknowledge J. Zimmerman and A.C. Gossard for the growth of the AlGaAs/GaAs heterostructure, F. Kuemmeth for useful discussions and Bobak Shahriari for proof reading the paper. This work was supported by the Royal Society, the EPSRC National Quantum Technology Hub in Networked Quantum Information Technology (EP/M013243/1), Quantum Technology Capital (EP/N014995/1), EPSRC Platform Grant (EP/R029229/1), the European Research Council (Grant agreement 818751), the Swiss NSF Project 179024, the Swiss Nanoscience Institute, the NCCR QSIT and the EU H2020 European Microkelvin Platform EMP grant No. 824109. This publication was also made possible through support from Templeton World Charity Foundation and John Templeton Foundation. The opinions expressed in this publication are those of the authors and do not necessarily reflect the views of the Templeton Foundations.

## Author contributions
D.T.L., N.E., F.V., and N.A. and the machine performed the experiments. H.M. developed the algorithm in collaboration with J.K., M.A.O., and D.S. The sample was fabricated by L.C.C., L.Y., and D.M.Z. The project was conceived by G.A.D.B., E.A.L., and N.A.

## Competing interests
The authors declare no competing interests.
