## [Peer Review File · Nature Communications]

Reviewers' comments:

Reviewer #1 (Remarks to the Author):

The manuscript by Moon et al. presents experimental tests of an autotuning algorithm that takes a GaAs-based gate-defined QD device with 8 gate electrodes and tunes the device into a double quantum dot regime in a little over an hour. The same success was achieved in another DQD device, and the authors investigated different aspects of their tuning algorithm and evaluates their relative contribution to the tuning success.

This is nice and detailed experimental work with immediate relevance to the QD community. Over the recent years, gate-defined quantum dots have been gradually scaling up beyond single-qubit regimes and the complexity of device tuning correspondingly grows at a very fast rate. Efforts to automate the dot tuning process has naturally become a technical goal of the community, with quite a few papers from several groups in this area. On the hand, due to the presence of disorder in QD devices, the large device-to-device variability has precluded fully automated tuning of DQD from "scratch" (i.e. human intervention is usually needed to tune the device into a vicinity of a working regime before the computer takes over). This is, to my knowledge, one of the first works that shows fully automated tuning from a blank slate. Although there are limitations to the authors' work (the stability diagram out of the automated tuning seems rather suboptimal to my eye), the full automation is an important feature of the authors' work that warrants a publication in Nat. Comm.

It would be nice to address a couple of questions before publication:

1. Why haven't the authors utilized the QPC that seems to be present on their device? Transport measurements are largely considered to be very coarse and have limited functionality beyond getting the device roughly where it should be. Charge sensor measurements are generally much cleaner. Maybe the authors should clarify why they have chosen transport measurements, and evaluate possible extension of their algorithm to QPC detection.
2. Is the stability diagram in Fig. 2 achieved using auto-tuning or human tuning? Maybe I missed it, but it's important to clarify if it is achieved by human tuning as it is significantly nicer than the plots from machine tuning.

Reviewer #2 (Remarks to the Author):

The paper describes a method for tuning quantum dot devices in an up to eight-dimensional gate voltage space in under 70 minutes, exceeding a human benchmark.

The proposed approach is claimed to be readily applicable to different material systems and device architectures.

The proposed method can give a quantitative measurement of device variability.

The work includes an ablation study to identify the benefits of each of its individual components.

The proposed method includes two key phases. The first one is the sampling phase, where the

algorithm aims at identifying where transport features are likely to be found on a hypersurface that separates low and high current regions in parameter space.

Overall, I found the paper hard to read and too confusing in some parts. I encourage the authors to revise the document to improve its clarity as much as possible.

I would explain better what a pinched-off region is and why it originates.

The pruning operation is hard to understand. Its motivation and clarity should be improved.

What does it mean "in the vicinity of a given v "?

I found the plots in Figure 3 hard to understand, especially part b). What is the green circle in this plot?

The motivation for the peak detection steps should be more clear. The same for the Score function decision step.

Why are V_3 and V_7 chosen for the peak detection step?

What happens if the high-resolution current map is checked by a human and the human reports that the device is not tuned to the double quantum dot regime?

Is this step, where a human has to check the output of the algorithm strictly necessary? If so, the proposed approach is not fully automated, which is undesirable.

I am concerned about the lack of reproducibility of the work due to the high complexity of the proposed algorithm, the many different parts and the lack of full details.

Will the authors provide any source code implementing their algorithm for reproducibility?

I believe the authors should address the previous comments before the paper can be accepted for publication.

Reviewer #3 (Remarks to the Author):

This paper presents a machine learning (ML) algorithm that is able to automatically tune quantum dot devices made from electrostatically gated GaAs heterostructures. The algorithm uses a module ('Sampling Phase') that determines a 'hypersurface' in the space of 8 gate voltages ($V_1, V_2 \dots V_8$) that define the quantum dot device. The hypersurface is the region in gate-voltage space where the threshold for current flow occurs. It is in this vicinity where single electron tunnelling through the quantum dots occurs. A second module in the algorithm ('Investigation Phase') then searches for Coulomb blockade (CB) oscillations, which occur near the transport threshold, in the vicinity of the hypersurface. Regular CB oscillations are a clear signature that a well-defined quantum dot has been formed.

The paper demonstrates that the designed algorithm performs much better than a 'random' selection of gate voltages, followed by the search for oscillations, which is what one would hope and expect, showing that this algorithm is working efficiently.

The paper also demonstrates that the algorithm can find 'double quantum dot' transport features in an

average time of 2.8 hrs for Device 1 and 1.1 hrs for Device 2. These times are achieved without any a priori knowledge of the 'function' or purpose of the gate electrodes. When one uses the 'design knowledge' that certain gates perform similar functions, the algorithm can then perform device tuning into the double-dot regime in 0.6 hrs.

The authors make the point (based on a private communication with Ferdinand Kuemmeth, and others) that the typical time for humans to tune up such devices is around 3 hours, thus justifying the main claim of their paper that this machine learning algorithm can outperform 'human experts'.

In my opinion the algorithm presented here is novel, although I note that there have been a number of recent papers all exploring different approaches to automatically tune quantum dot devices, some of which also involve machine learning. Many of these papers are indeed referenced in this current submission.

I also believe that the analysis of the algorithm's performance is sound and has been done carefully. The methodology is also well explained in the text, and the appendices.

Despite this, I am not convinced that this work reaches the level of significance and impact needed to justify publication in Nature Communications. One reason for my judgment is that the quality of the double quantum dot stability maps obtained (in Figs. 4 & 5) is generally quite poor. This could simply be because there was considerable disorder in the devices themselves, but I find this surprising, since these were GaAs quantum dots, which are generally very low disorder and now routinely show high-quality multiple quantum dot behavior.

Clear double-dot stability maps for GaAs devices are now routinely demonstrated by many groups worldwide. For example, if one refers to the stability maps produced by the auto-tuning algorithms presented in Ref. [12] – Mills et al., APL 115, 113501 (2019), or Ref. [13] – Zwolak et al., arXiv: 1909.08030, these are very clearly due to well-defined double quantum dots. In contrast, while some of the features in Figs. 4 & 5 of the submitted manuscript do appear to be related to double-dot-like potentials, it is not at all clear whether these are just due to random disorder in the samples, or if they are the intended dots as defined by the gate electrodes.

I think this paper would have more impact in the field if the stability maps generated by the ML algorithm were more convincingly those intended by the device design. One suggestion is that the authors try some other devices which show clear double-dot stability maps (perhaps when generated by human control), and then try out the algorithm on those.

The paper as it stands, however, is clearly publishable, but I would suggest it was more suitable to a more specialized journal, perhaps IoP's Quantum Science & Technology?

Answers to Reviewer 1

The manuscript by Moon et al. presents experimental tests of an autotuning algorithm that takes a GaAs-based gate-defined QD device with 8 gate electrodes and tunes the device into a double quantum dot regime in a little over an hour. The same success was achieved in another DQD device, and the authors investigated different aspects of their tuning algorithm and evaluates their relative contribution to the tuning success.

This is nice and detailed experimental work with immediate relevance to the QD community. Over the recent years, gate-defined quantum dots have been gradually scaling up beyond single-qubit regimes and the complexity of device tuning correspondingly grows at a very fast rate. Efforts to automate the dot tuning process has naturally become a technical goal of the community, with quite a few papers from several groups in this area. On the hand, due to the presence of disorder in QD devices, the large device-to-device variability has precluded fully automated tuning of DQD from “scratch” (i.e. human intervention is usually needed to tune the device into a vicinity of a working regime before the computer takes over). This is, to my knowledge, one of the first works that shows fully automated tuning from a blank slate. Although there are limitations to the authors’ work (the stability diagram out of the automated tuning seems rather suboptimal to my eye), the full automation is an important feature of the authors’ work that warrants a publication in Nat. Comm.

It would be nice to address a couple of questions before publication: 1. Why haven’t the authors utilized the QPC that seems to be present on their device? Transport measurements are largely considered to be very coarse and have limited functionality beyond getting the device roughly where it should be. Charge sensor measurements are generally much cleaner. Maybe the authors should clarify why they have chosen transport measurements, and evaluate possible extension of their algorithm to QPC detection.

2. Is the stability diagram in Fig. 2 achieved using auto-tuning or human tuning? Maybe I missed it, but it’s important to clarify if it is achieved by human tuning as it is significantly nicer than the plots from machine tuning.

Answer to R1.0 We thank the reviewer for recommending publication. Below we address the questions raised.

Comment R1.1 *Why haven’t the authors utilized the QPC that seems to be present on their device? Transport measurements are largely considered to be very coarse and have limited functionality beyond getting the device roughly where it should be. Charge sensor measurements are generally much cleaner. Maybe the authors should clarify why they have chosen transport measurements, and evaluate possible extension of their algorithm to QPC detection.*

Answer to R1.1 There are two reasons not to use a charge sensor in this experiment. Firstly, the sensor must itself be tuned to its optimum operating configuration, which depends on the other gate voltages applied to the device. To automate this process adds significant complexity, which is beyond the scope of this paper (although we are working on it). We have added a sentence to the conclusion mentioning the automation of charge sensing techniques as an outlook. Secondly, a charge sensing QPC usually works in only part of the gate voltage parameter space. Our algorithm explores the entire parameter space and thus relies on transport measurements, which are always available.

Comment R1.2 *Is the stability diagram in Fig. 2 achieved using auto-tuning or human tuning? Maybe I missed it, but it's important to clarify if it is achieved by human tuning as it is significantly nicer than the plots from machine tuning.*

Answer to R1.2 This stability diagram was produced using the auto-tuning algorithm; we have added this information to the caption. The reason why this stability diagram is not included in the results (Figure 5 and 6 of the revised manuscript) is that we run the algorithm only once in that particular thermal cycle of Device 2 due to difficulties with the dilution refrigerator, so we could not perform an statistical analysis of the algorithm's performance. We now use this algorithm routinely in our laboratory to tune every device, as it is much quicker than doing it manually.

Answers to Reviewer 2

The paper describes a method for tuning quantum dot devices in an up to eight-dimensional gate voltage space in under 70 minutes, exceeding a human benchmark.

The proposed approach is claimed to be readily applicable to different material systems and device architectures.

The proposed method can give a quantitative measurement of device variability.

The work includes an ablation study to identify the benefits of each of its individual components.

The proposed method includes two key phases. The first one is the sampling phase, where the algorithm aims at identifying where transport features are likely to be found on a hypersurface that separates low and high current regions in parameter space.

Overall, I found the paper hard to read and too confusing in some parts. I encourage the authors to revise the document to improve its clarity as much as possible.

I would explain better what a pinched-off region is and why it originates.

The pruning operation is hard to understand. Its motivation and clarity should be improved.

What does it mean "in the vicinity of a given v "?

I found the plots in Figure 3 hard to understand, especially part b). What is the green circle in this plot?

The motivation for the peak detection steps should be more clear. The same for the Score function decision step.

Why are $V3$ and $V7$ chosen for the peak detection step?

What happens if the high-resolution current map is checked by a human and the human reports that the device is not tuned to the double quantum dot regime?

Is this step, where a human has to check the output of the algorithm strictly necessary? If so, the proposed approach is not fully automated, which is undesirable.

I am concerned about the lack of reproducibility of the work due to the high complexity of the proposed algorithm, the many different parts and the lack of full details.

Will the authors provide any source code implementing their algorithm for reproducibility?

I believe the authors should address the previous comments before the paper can be accepted for publication.

Comment R2.1 *Overall, I found the paper hard to read and too confusing in some parts. I encourage the authors to revise the document to improve its clarity as much as possible. I would explain better what a pinched-off region is and why it originates. The pruning operation is hard to understand. Its motivation and clarity should be improved. What does it mean "in the vicinity of a given v "? I found the plots in Figure 3 hard to understand, especially part b). What is the green circle in this plot? The motivation for the peak detection steps should be more clear. The same for the Score function decision step.*

Answer to R2.1 To improve the manuscript's clarity, we have replaced sections I and II with a section called 'Description of the algorithm', which is written from scratch. We have expanded the algorithm flow diagram into a new clearer figure (formerly Fig. 2b, now Fig. 3). We have heavily revised Fig. 3 (now Fig. 4) and completely rewritten the caption. We have revised several paragraphs in the rest of the manuscript. We have created animations to illustrate how the algorithm works (displayed in the compressed file for the refactored algorithm, AutoDot_refactored.zip, path: AutoDot/Resources/Algorithm_overview/README.html). We have also expanded on the explanation of the peak detection and score function steps, and we have added a new section in

the Supplementary Material (‘Overview of the algorithm’) with a mathematical justification of the algorithm’s components. We recognise that a pinched-off region is jargon and have defined this concept in the manuscript. We have also clarified the motivation of the pruning operation. By the “vicinity of a given v ”, we mean the coordinates that are close to v in gate voltage space. We now explain quantitatively what this means (Section IB, paragraph 3).

Comment R2.2 *Why are V_3 and V_7 chosen for the peak detection step?*

Answer to R2.2 The device geometry means that gate voltages V_3 and V_7 couple most strongly to the quantum dot energy levels and weakly to the tunnel barriers. (In the jargon of the field, they are the “plungers”.) These are the voltages that a human would usually sweep to identify double quantum dot behaviour in a two-dimensional current map. We therefore designed the algorithm to look for transport features in this same measurement sweep. The users can define which gate electrodes are chosen to act as “plungers” according to their particular device geometry. We have clarified this point in the manuscript in Section IB.

Comment R2.3 *What happens if the high-resolution current map is checked by a human and the human reports that the device is not tuned to the double quantum dot regime? Is this step, where a human has to check the output of the algorithm strictly necessary? If so, the proposed approach is not fully automated, which is undesirable.*

Answer to R2.3 Human checking is only necessary to evaluate the performance of the algorithm, but not during the algorithm’s operation. We developed this algorithm with the purpose of identifying promising locations in the high-dimensional gate voltage space. This output can be either inspected by a human (as here) or fed as inputs to another algorithm, for example for fine-tuning. We have clarified this in the paragraph 1 and 8 of Section II.

Comment R2.4 *I am concerned about the lack of reproducibility of the work due to the high complexity of the proposed algorithm, the many different parts and the lack of full details. Will the authors provide any source code implementing their algorithm for reproducibility?*

Answer to R2.4 Within our own laboratory, we have verified that the algorithm is reproducible by measuring two different devices and over different cycles from room temperature to operating temperature. This shows that the algorithm is robust against fabrication variation and against redistribution of trapped charges. To ensure reproducibility in other laboratories, we have made the implementation available in two versions as described in Answer AE.1 above.

We agree that the algorithm is complex and has many parts, but the ablation study in Section II B shows that all these parts are necessary. We have also included a mathematical justification of the algorithm’s components in the section ‘Overview of the algorithm’ in the Supplementary Material. The beauty of our approach is that it exploits the fact that the physics of these devices is similar and the parameter space has common features, while at the same time it is able to cope with fabrication differences and material defects that make the tuning difficult without automation.

In response to the reviewer’s earlier comments, we have added more details of the experiment as described above.

Answers to Reviewer 3

This paper presents a machine learning (ML) algorithm that is able to automatically tune quantum dot devices made from electrostatically gated GaAs heterostructures. The algorithm uses a module ('Sampling Phase') that determines a 'hypersurface' in the space of 8 gate voltages ($V_1, V_2 \dots V_8$) that define the quantum dot device. The hypersurface is the region in gate-voltage space where the threshold for current flow occurs. It is in this vicinity where single electron tunnelling through the quantum dots occurs. A second module in the algorithm ('Investigation Phase') then searches for Coulomb blockade (CB) oscillations, which occur near the transport threshold, in the vicinity of the hypersurface. Regular CB oscillations are a clear signature that a well-defined quantum dot has been formed.

The paper demonstrates that the designed algorithm performs much better than a ?random? selection of gate voltages, followed by the search for oscillations, which is what one would hope and expect, showing that this algorithm is working efficiently.

The paper also demonstrates that the algorithm can find 'double quantum dot' transport features in an average time of 2.8 hrs for Device 1 and 1.1 hrs for Device 2. These times are achieved without any a priori knowledge of the 'function' or purpose of the gate electrodes. When one uses the 'design knowledge' that certain gates perform similar functions, the algorithm can then perform device tuning into the double-dot regime in 0.6 hrs.

The authors make the point (based on a private communication with Ferdinand Kuemmeth, and others) that the typical time for humans to tune up such devices is around 3 hours, thus justifying the main claim of their paper that this machine learning algorithm can outperform 'human experts'. In my opinion the algorithm presented here is novel, although I note that there have been a number of recent papers all exploring different approaches to automatically tune quantum dot devices, some of which also involve machine learning. Many of these papers are indeed referenced in this current submission.

I also believe that the analysis of the algorithm's performance is sound and has been done carefully. The methodology is also well explained in the text, and the appendices.

Despite this, I am not convinced that this work reaches the level of significance and impact needed to justify publication in Nature Communications. One reason for my judgment is that the quality of the double quantum dot stability maps obtained (in Figs. 4 & 5) is generally quite poor. This could simply be because there was considerable disorder in the devices themselves, but I find this surprising, since these were GaAs quantum dots, which are generally very low disorder and now routinely show high-quality multiple quantum dot behavior.

Clear double-dot stability maps for GaAs devices are now routinely demonstrated by many groups worldwide. For example, if one refers to the stability maps produced by the auto-tuning algorithms presented in Ref. [12] Mills et al., APL 115, 113501 (2019), or Ref. [13] Zwolak et al., arXiv: 1909.08030, these are very clearly due to well-defined double quantum dots. In contrast, while some of the features in Figs. 4 & 5 of the submitted manuscript do appear to be related to double-dot-like potentials, it is not at all clear whether these are just due to random disorder in the samples, or if they are the intended dots as defined by the gate electrodes.

I think this paper would have more impact in the field if the stability maps generated by the ML algorithm were more convincingly those intended by the device design. One suggestion is that the authors try some other devices which show clear double-dot stability maps (perhaps when generated by human control), and then try out the algorithm on those.

The paper as it stands, however, is clearly publishable, but I would suggest it was more suitable to a more specialized journal, perhaps IoP's Quantum Science & Technology?

Comment R3.1 *In my opinion the algorithm presented here is novel [...]. I also believe that the analysis of the algorithm’s performance is sound and has been done carefully. The methodology is also well explained in the text, and the appendices.*

Answer to R3.1 We thank the reviewer for recognising the novelty and soundness of our approach and the clarity of the methodology.

Comment R3.2 *Despite this, I am not convinced that this work reaches the level of significance and impact needed to justify publication in Nature Communications. One reason for my judgment is that the quality of the double quantum dot stability maps obtained (in Figs. 4 and 5) is generally quite poor. This could simply be because there was considerable disorder in the devices themselves, but I find this surprising, since these were GaAs quantum dots, which are generally very low disorder and now routinely show high-quality multiple quantum dot behavior. Clear double-dot stability maps for GaAs devices are now routinely demonstrated by many groups worldwide. For example, if one refers to the stability maps produced by the auto-tuning algorithms presented in Ref. [12] Mills et al., APL 115, 113501 (2019), or Ref. [13] Zwolak et al., arXiv: 1909.08030, these are very clearly due to well-defined double quantum dots. In contrast, while some of the features in Figs. 4 and 5 of the submitted manuscript do appear to be related to double-dot-like potentials, it is not at all clear whether these are just due to random disorder in the samples, or if they are the intended dots as defined by the gate electrodes. I think this paper would have more impact in the field if the stability maps generated by the ML algorithm were more convincingly those intended by the device design. One suggestion is that the authors try some other devices which show clear double-dot stability maps (perhaps when generated by human control), and then try out the algorithm on those.*

Answer to R3.2 Unlike Ref [12] and [13], our algorithm is able to start from scratch, and thus only direct transport measurements are possible. As we have clarified in our answer R1.1 above, other readout methods are not always available and require the sensor to be tuned to operating conditions at the different regions of gate voltage space explored. Although charge sensor measurements (as those in Ref [12] and [13]) are generally much cleaner, as pointed out by reviewer 1, these are not suitable for our much coarser tuning approach. We have explained this in paragraph 4 of the Introduction.

A strength of our algorithm is that it can perform well in devices which are not perfect and in which the double-dot features are not as evident as in the cleanest devices. We believe the impact of our work is much greater if we show that our algorithm performs well in devices that are not perfect or ‘Hollywood like’. We want to show an algorithm that can be applied in every lab and for every device. We emphasise that we could not find clearer double quantum dot stability diagrams by manual tuning the devices. We now use the algorithm routinely in our laboratory to tune double quantum dots, since we know it is faster and more thorough than we are in that search.

Another important point is that other approaches can be used to improve on the stability diagrams found, i.e. to fine tune it, once the device is coarse tuned. Another of our papers, arXiv:2001.04409, shows how this is feasible completely automatically. We have now explained this on page 7 of the manuscript. Overall, this work focuses on the extremely challenging task of coarse tuning devices which are not necessarily perfect, by efficiently navigating a huge parameter space. Our algorithm is gate-architecture-agnostic and designed to be robust in different scenarios of disorder and other defects.

REVIEWERS' COMMENTS:

Reviewer #1 (Remarks to the Author):

The authors have satisfactorily addressed my comments and I recommend publication.

Reviewer #2 (Remarks to the Author):

I have looked at the updated version of the paper and I think the authors have successfully addressed my concerns. I believe the new detailed description of the proposed algorithm, together with figures 3 and 4, is very informative.

Reviewer #3 (Remarks to the Author):

In this revised submission the authors have worked hard to further explain the details of the ML algorithm and how it works, as requested by Reviewer 2. I found the original version relatively clear, and the extra information now provided makes it even clearer. In general, I have no concerns about the explanations of the algorithm, or indeed its validity.

The main concern that I noted in my previous review was that it was not clear to me that the algorithm was reliably able to tune the devices into a double-dot bias regime that would be useful for undertaking spin qubit operations, which is the motivation for this work. The issue as I saw it was that the quality of the double dot stability maps shown in Figs. 4 & 5 is quite poor, and so it was not clear to me that these bias conditions were really the ones that a researcher would want to use for qubit measurements.

In their response letter (in response to a question by Reviewer 1), and in the revised manuscript, the authors have pointed out that the double-dot stability map data shown in Fig. 1b (& Fig. 2 of revised manuscript) were, in fact, obtained using their ML algorithm. I had not realised this earlier, and I thank Reviewer 1 for asking this question. This stability does indeed appear to be what one would expect for the device structure shown in Figure 1a, and provides confidence that the algorithm is able to (on some occasions at least) configure the gate biases into a region where spin qubit experiments could be further attempted.

Furthermore, in their response to my concern, the authors have pointed out the very important fact that they have also recently posted another paper (arXiv:2001.04409, Ref. [21]) which describes an additional ML algorithm that they have developed, which goes beyond the initial 'coarse biasing' described in the present manuscript, and which enables 'fine-tuning' of the biases into a regime which is better suited for spin qubit measurements.

I've now taken a look at their related paper (arXiv:2001.04409), and it seems like this second ML algorithm, when combined with the one described in the present submission, could work well together, and could produce biasing conditions suitable for studying spin effects, such as spin blockade, in arbitrarily configured quantum dot devices.

In an ideal world, it would be nice if the two papers were combined into one coordinated story, so that one could see the 'complete picture' of going from scratch to a well-tuned double dot. However, I do appreciate that this would be a very long paper, and the two algorithms were likely developed by different researchers in the Oxford group.

Another concern that I have with this technique (and I recognise that I didn't raise this in my first review), is that it is designed purely for electrical transport measurements, rather than for charge sensing. This is an issue because it is very difficult to demonstrate spin qubits (in quantum dots) with only transport-based sensing, which is why most groups studying spin qubits use either a QPC or SET charge sensor, or gate-based dispersive readout. As a result, I see it as unlikely that the algorithm described here, which is based around finding 'current thresholds' or 'pinch off' points, will gain much utility in future spin qubit experiments. I do agree with the authors, though, that it could be useful to assess the 'variability' of test devices, in order to assess new fabrication processes, for example.

Overall, I'm still not sure that this paper reaches the level of impact commensurate with Nature Communications. Having said that, I do think it is an excellent paper, and many in the quantum device measurement community will find it interesting, primarily because it demonstrates an automated way to find threshold voltages 'from scratch'.

Answer to reviewer comments

Reviewer #1:

The authors have satisfactorily addressed my comments and I recommend publication.

Answer to reviewer #1:

We thank reviewer 1 for his/her support.

Reviewer #2:

I have looked at the updated version of the paper and I think the authors have successfully addressed my concerns. I believe the new detailed description of the proposed algorithm, together with figures 3 and 4, is very informative.

Answer to reviewer #2:

We are grateful to reviewer 2 for helping us strengthen our manuscript.

Reviewer #3:

In this revised submission the authors have worked hard to further explain the details of the ML algorithm and how it works, as requested by Reviewer 2. I found the original version relatively clear, and the extra information now provided makes it even clearer. In general, I have no concerns about the explanations of the algorithm, or indeed its validity.

The main concern that I noted in my previous review was that it was not clear to me that the algorithm was reliably able to tune the devices into a double-dot bias regime that would be useful for undertaking spin qubit operations, which is the motivation for this work. The issue as I saw it was that the quality of the double dot stability maps shown in Figs. 4 & 5 is quite poor, and so it was not clear to me that these bias conditions were really the ones that a researcher would want to use for qubit measurements.

In their response letter (in response to a question by Reviewer 1), and in the revised manuscript, the authors have pointed out that the double-dot stability map data shown in Fig. 1b (& Fig. 2 of revised manuscript) were, in fact, obtained using their ML algorithm. I had not realised this earlier, and I thank Reviewer 1 for asking this question. This stability does indeed appear to be what one would expect for the device structure shown in Figure 1a, and provides confidence that the algorithm is able to (on some occasions at least) configure the gate biases into a region where spin qubit experiments could be further attempted.

Furthermore, in their response to my concern, the authors have pointed out the very important fact that they have also recently posted another paper

(arXiv:2001.04409, Ref. [21]) which describes an additional ML algorithm that they have developed, which goes beyond the initial ‘coarse biasing’ described in the present manuscript, and which enables ‘fine-tuning’ of the biases into a regime which is better suited for spin qubit measurements.

I’ve now taken a look at their related paper (arXiv:2001.04409), and it seems like this second ML algorithm, when combined with the one described in the present submission, could work well together, and could produce biasing conditions suitable for studying spin effects, such as spin blockade, in arbitrarily configured quantum dot devices.

In an ideal world, it would be nice if the two papers were combined into one coordinated story, so that one could see the ‘complete picture’ of going from scratch to a well-tuned double dot. However, I do appreciate that this would be a very long paper, and the two algorithms were likely developed by different researchers in the Oxford group.

Another concern that I have with this technique (and I recognise that I didn’t raise this in my first review), is that it is designed purely for electrical transport measurements, rather than for charge sensing. This is an issue because it is very difficult to demonstrate spin qubits (in quantum dots) with only transport-based sensing, which is why most groups studying spin qubits use either a QPC or SET charge sensor, or gate-based dispersive readout. As a result, I see it as unlikely that the algorithm described here, which is based around finding ‘current thresholds’ or ‘pinch off’ points, will gain much utility in future spin qubit experiments. I do agree with the authors, though, that it could be useful to assess the ‘variability’ of test devices, in order to assess new fabrication processes, for example.

Overall, I’m still not sure that this paper reaches the level of impact commensurate with Nature Communications. Having said that, I do think it is an excellent paper, and many in the quantum device measurement community will find it interesting, primarily because it demonstrates an automated way to find threshold voltages ‘from scratch’.

Answer to reviewer #3: We thank reviewer 3 for acknowledging that our algorithm is, in some occasions, capable of configuring gate biases into a region where spin qubit experiments could be directly attempted. In other occasions, fine tuning algorithm such as the one described in arXiv:2001.04409 would be necessary. We agree with the reviewer that a paper with fine and coarse tuning combined would allow the reader to see the complete picture, but as the reviewer points out, this would result in a very

long and difficult to read paper, and indeed, these two algorithms were developed by different people in the group.

The other concern that the reviewer mentions, that our algorithm is designed purely for electrical transport measurements rather than for charge sensing, was also touched upon by reviewer 1, and it has been addressed in the manuscript by adding the following sentences:

- in the Introduction:

‘Although other techniques for measuring the double quantum dot exist, such as charge sensing and dispersive readout, they also require other parameters to be re-tuned when the gate voltages vary and are therefore not suitable for automated measurements.’

- in the Discussion:

‘The use of charge sensors and RF readout could also be implemented to improve tuning times, although these techniques would require to be automatically tuned to their optimum operating configuration, and would be restricted to small regions of the gate voltage space.’